# Compressed Brown Algae as a Potential Environmental Enrichment Material in Growing Pigs

**DOI:** 10.3390/ani11020315

**Published:** 2021-01-27

**Authors:** Françoise Pol, Adeline Huneau-Salaün, Sarah Gallien, Yannick Ramonet, Nicolas Rose

**Affiliations:** 1Ploufragan-Plouzané-Niort Laboratory, EPISABE Unit, ANSES, B.P. 53, 22440 Ploufragan, France; adeline.huneau@anses.fr (A.H.-S.); sarah.gallien1991@gmail.com (S.G.); nicolas.rose@anses.fr (N.R.); 2Chambre Régionale D’agriculture de Bretagne, Pôle Porc, 22195 Plérin, France; yannick.ramonet@bretagne.chambagri.fr

**Keywords:** welfare, seaweed, manipulable material, housing enrichment

## Abstract

**Simple Summary:**

To prevent pigs from redirecting their exploratory natural behavior to their penmates, so as to prevent tail biting and promote pig welfare, the Council Directive 2008/120/EC has made environmental enrichment in pig farming mandatory. Possible enrichments can be varied: objects that are edible or not or bulk material. However, the enrichments need to be compatible with slatted floors, the main type of floor in pig housing. Here, we assessed the impact on pig welfare of a material made of an algae-based cylinder, compatible with slatted floors and edible, and which is categorized as suboptimal enrichment materials according to Commission Recommendation (EU) 2016/336 classification. The assessment was done in comparison with metal chains, an enrichment material commonly used in farms, and with wood logs, a bio-sourced object too. Enrichment material made from algae had no negative effect on pig health. Performance and body condition were similar between groups that were given different materials. When provided during the suckling period, the algae material did not appear to promote manipulating behavior in piglets after weaning. Regarding their characteristics, the algae cylinders can be categorized as suboptimal enrichment materials, although it does not significantly improve pig welfare when compared to a metal chain, which is categorized as a material of marginal interest.

**Abstract:**

In barren housing conditions, enrichment materials are given to pigs to improve their welfare. Here, we assessed the suitability of an algae-based cylinder as an enrichment material on the behavioral, physiological, health and productivity welfare indicators of pigs. Algae was compared with metal chains and wood logs. The study involved 444 pigs from two successive batches on one single farm. During the suckling period, half of the pigs received algae and the control pigs received no material. After weaning and until the end of fattening, algae, wood or chains were equally distributed among the pigs. Consumption of algae cylinders was different between pens and between batches. After weaning and during the fattening period, although the results differed between batches, no significant difference was observed in the object manipulations. Salivary cortisol, used as biomarker to measure the stress levels after pig transfers, were similar between the treatments. Enrichment material made from algae had no negative effect on pig health and no effect on performance and body condition. Regarding their characteristics and according to Commission Recommendation (EU) 2016/336 classification, algae cylinders can be categorized as suboptimal enrichment materials, although the present results suggest that it does not significantly improve pig welfare compared to a metal chain, which is categorized as a material of marginal interest.

## 1. Introduction

Pigs have a natural tendency to explore the environment in which they live, searching for, foraging and chewing on any materials they can find. However, in commercial rearing conditions, pigs’ environments are generally barren and do not provide adequate substrate to express these behaviors. To prevent pigs from redirecting their exploratory behavior to their congeners, so as to prevent tail biting, which is a major welfare concern in pig farms, the Council Directive 2008/120/EC promotes environmental enrichment [1]. More recently, a recommendation by the Commission [2] and a staff working document [3] gave a description of adequate enrichment materials.

However, most pigs in the European Union (EU) are housed on partly or fully slatted floors [4] and finding an adequate enrichment material for pigs housed on this type of floor is not straightforward. Straw and other rootable materials, such as roughage, which are considered as optimal materials when given as bedding materials or as a sub-optimal materials when given in racks [3], are not suitable because they can block the slats and thus the slurry handling system [5]. Enrichment objects are more suitable for pigs reared on slatted floors. The most popular is a suspended metal chain, which is commonly used in France and other EU countries [6]. However, this object is not bio-sourced, nor edible or destructible, and is thus considered of marginal interest according to the EU staff working document [3]. Since February 2020, a metal chain can no longer be used on its own in France [7]. Wood is chewable and destructible and is more likely to sustain the interest of pigs [8,9,10,11,12,13]. However, the suitability of wood logs or blocks as an enrichment material depends on the wood type. Hard wood is considered of marginal interest because it may splinter when bitten into [3,14]. To be attractive, the wood has to be fresh and odorous [3], which complicates procurement and may cause biosecurity trouble depending on the wood origin. Other bio-sourced objects can be given to pigs to improve their welfare. Sisal rope arouses pig interest [15,16] more than a metal chain [17] and decreases penmate manipulation [18]. Hessian sack, also easy to bite and shake, commonly used for sows before farrowing, are also well used by young pigs and reduce damaging behaviors [19,20]. Wooden briquettes given to weaned pigs did not give better results on welfare indicators than a metal chain [21].

The aim of our study was to test the enriching properties of a new bio-sourced object made of brown algae, which gathers most of the qualities defined by the EU Recommendation and which can be considered as sub-optimal material. It is a potential by-product of the cosmetics or food-processing industry, so it is cheap and available in large quantities; it is edible and nutrients can be incorporated into it; it is also safe and hygienic due to a high heat treatment during the manufacturing process. Furthermore, its manufacturing process can be standardized and developed at an industrial scale. During a pre-trial on few animals, the attractiveness and the safety of the material for the pigs were checked. The objective of this study, conducted on an experimental farm, was therefore to test the algae material as an enrichment object given to growing pigs from 15 days of life to one month before slaughter. A set of behavioral and physiological indicators, animal health, body condition and zootechnical performances was used to assess the welfare of the animals. This assessment was made in comparison with other objects suitable for slatted floors, specifically, metal chains, which are widely used on commercial farms, and wood logs, another bio-sourced material used on farms, to evaluate the potential advantage of this new material on pigs’ behavior, health and performance.

## 2. Materials and Methods

### 2.1. Animals and Housing Conditions

The experiment was done in the experimental facilities of Crécom, Saint-Nicolas du Pélem, France. As none of the interventions were likely to cause pain or distress to the pigs, no ethics committee approval was needed. The Large White × Landrace × Pietrain pigs were raised in accordance with the requirements of Council Directive 2008/120/EC [1].

Two successive batches of pigs (B1 and B2) composed of 20 litters each were studied from 15 days old (D15) to 104 days old (D104). The sex ratio was 121/107 for B1 and 104/110 for B2. Pigs from each batch were housed in two farrowing rooms of 10 pens during the suckling period. Each sow was housed in an individual farrowing stall and the piglets could circulate in the pen around her. The piglets were then transferred to two rooms of 10 pens after weaning at D28. One litter was housed per pen and, according to litter size, each pig had between 0.48 and 0.62 m^2^ available floor area. The temperature in the nursery was 27 °C when the animals arrived, which was slowly decreased to reach 21 °C, 21 days after (D49), and the air volume available for each pig was 1.0 m^3^. At D61, growers were transferred to two fattening rooms, one with 8 pens and one with 12 pens. Eight to thirteen pigs were housed in each pen, depending on the litter and pen size, so that each pig had between 0.69 and 1.27 m^2^ available floor area. The temperature of the rooms was 23 °C and the air volume available for each pig was 1.4 m^3^.

Throughout their lives, pigs received commercial feed with an energy and a nutrient supply in line with INRA recommendations [22,23]. Both batches received dry feed during nursery and, during the growing period, batch B1 received dry feed and batch B2 received liquid feed. All pigs were tail-docked and teeth-clipped within the first three days of life, as commonly practiced in French commercial farms upon veterinary advice for this specific farm. The males were castrated within the first week of life. The floors of all pens were fully slatted.

### 2.2. Enrichment Materials

Three different objects were used during the trial. The first material was an algae material (Algopack, Saint-Malo, France) made of *Saccharina latissimi*, which is a brown macroalgae or seaweed, post-harvest dried and compressed under heat into a cylinder (15 × 5 cm, 380 g) with an axial hole (Appendix A and Figure 1a). From D15 to D28, during the suckling period, the cylinder was placed horizontally on a roll dispenser mounted on the floor (height of dispenser: 12 cm above the floor) located 50 cm from the sow’s head (Figure 1b). The cylinder can spin around its axis on the roll dispenser. One cylinder was placed in each pen. After weaning, in the post-weaning and fattening rooms and until D104, the cylinders were threaded into the axis of metal vertical roll dispensers (65 × 7 cm). Only the bottom of the cylinder was accessible to the pigs (Figure 1c). The dispenser was mounted on the pen wall at a height of 12 cm in the post-weaning pens and at 30 cm in the finishing pens. There was one dispenser in each pen. The second material was wood beams (square section 5 × 5 cm, 40 cm, 300 g), which were supplied in the same device as the algae cylinders. The third material was metal chains made of 2.5 × 4 cm links (link section: 0.5 cm) suspended along the pen wall; the end of the chain was a few centimeters above the floor. Wood and metal chains were given from D28 to D104.

### 2.3. Experimental Scheme

All pigs of the same litter received the same treatment. They remained together in the same group during the whole trial. From D15 to D28, suckling pigs were divided into two groups of 10 litters. One group (Treatment A) was furnished with algae. The other group (Treatment C) received no enrichment (the control group). At D28, weaned pigs in Treatment A were subdivided into three treatments according to the enrichment object: algae (Treatment AA), wood (Treatment AW) or metal chain (Treatment AMC). Control pigs, which did not receive any enrichment materials during the suckling period, were also divided into the three enrichment treatments of algae, wood and metal chain (CA, CW and CMC, respectively) at D28. Each litter was assigned to one of the six treatments from after weaning until the end of the experiment. Treatments were balanced between rooms at each experimental step. The sex ratio was similar in all treatments. The number of pigs in each treatment is presented in Table 1.

### 2.4. Measurements

#### 2.4.1. Use of Enrichment Materials

During the trial, the dispensers were replenished with algae and wood cylinders. The wood blocks and algae cylinders were weighed at least weekly.

#### 2.4.2. Behavioral Observations

In nursery rooms, from D28 to D61, 16 pens per batch were video-recorded for 3.5 h in the morning and 3 h in the afternoon. Videos were shot on two consecutive days per week for five weeks. During one two-day session, the pigs were individually identified with a mark on their backs. Manipulation of the enrichment object was recorded by scan sampling every 10 min. After transfer to the growing pens, pigs were directly observed by two trained operators from 9:00 to 11:30 a.m. at D70, D75, D84 and D104. The operator moved slowly along the pens and recorded any manipulation of the material.

#### 2.4.3. Salivary Cortisol Level

Salivary cortisol levels were measured before and after the transfer from the farrowing pen to the nursery pen (in B2) and from the nursery pen to the finishing pen (in B1). Five pigs per litter were randomly selected and individually identified before transfer. For B2, the salivary samples were obtained by mouth-swabbing with a cotton swab at D28, D30 and D31 in the morning (from 8:00 to 9:00 a.m.); the swab, held with a metal hemostat, was shown to the pig, which then chewed on it. The first sampling point was before the transfer and the second and third points were, respectively, 24 h and 48 h after transfer. For B1, six pigs per pen were sampled at D61, D62 and D63. After sampling, swabs were placed in a dry tube (Salivette^®^, Sarstedt, Marnay, France), centrifuged (3500 rpm, 10 min) and frozen at −20 °C for storing before analysis. Salivary cortisol was assayed using a luminescence immunoassay (Cortisol Luminescence Immunoassay, IBL International, Hamburg, Germany). Concentrations were expressed as ng/mL of saliva.

#### 2.4.4. Body Conditions

Animal health was monitored daily. Lesion scoring was performed on all pigs at D15, the beginning of the experiment; at D28, D30, and D37, on the day before and the day after the first transfer and then one week after the first transfer; at D61 and D69, the day after the second transfer and one week after; and at D104, the end of the experiment. The lesion score was adapted from the Welfare Quality® project [24], with examination of the ears, the forequarters, the middle section of the body and the hindquarters, legs and tail. A three-level score was calculated according to the number of lesions observed: score 0 (slightly injured, less than 10 wounds on all body parts), score 1 (moderately injured, between 11 and 29 wounds on all body parts) and score 2 (heavily injured, more than 30 wounds in one part of the body or at least two parts with more than 20 wounds).

At D15, D28, D61 and D104, the pigs were individually weighed and the average daily gain (ADG) was calculated for the suckling period from D15 to D28, for the nursery period from D28 to D61 and for the growing period from D61 to D104. Feed intake was recorded at the half room level (included five pens as one unit) and the food conversion ratios were calculated. However, as the treatments were balanced between the rooms, these data were not available for each treatment.

### 2.5. Statistical Analysis

Raw comparisons between treatments were carried out using non-parametric tests based on the ranks (Wilcoxon test during the farrowing period and Kruskal–Wallis test during the nursery and growing periods) for algae consumption and the number of marked pigs manipulating the materials. Frequencies of lesions were compared with the χ^2^ test. Correlations between the material consumption at the three periods were calculated as the Spearman correlation coefficients based on the ranks. If a “batch” effect was detected, batches were analyzed separately. For multivariate modeling, enrichment material type (A vs. C from D15 to D28; A, W or MC from D29 to D104) was introduced as a fixed factor. A mixed ANOVA model with the pen as random effect was used to analyze, for observations on behavior (scans), the number of pigs manipulating the material and the individual weight, ADG, cortisol levels (R software, package geepack). For comparison purposes, behavioral counts in a pen were weighted to obtain counts on 10 pigs. Cortisol concentrations were log-transformed. Post-hoc tests were carried out using Tukey’s range tests. The risk of being injured (lesion) was modeled using a logistic regression model, introducing enrichment material and sex as the fixed effects; the litter effect was taken into account in a mixed model with the pen (a litter was housed in a single pen) as a random effect (package lme4). Repeated measures of lesions from D28 to D104 were taken into account with a repeated mixed regression model.

## 3. Results

### 3.1. Use of Enrichment Materials

Pigs manipulated and investigated the algae cylinder, leading to its slow degradation due to the action of the saliva. No residues of the cylinder were found on the floor or between the floor slats. Less than one cylinder was consumed per pen during the suckling period, from D15 to D28. The average daily consumption was 1.1 g/piglet (95% confidence interval (CI_95%_) 0.7–1.5) and strongly varied among pens from 0.1 to 2.4 g/day. The average consumption steadily increased to 1.4 ± 0.6 cylinders per pen during the nursery period, from D28 to D61, and to 3.5 ± 1.5 cylinders during the fattening period, from D61 to D104. Pen consumption of algae during the suckling period (treatment A) did not correlate with nursery period pen consumption (treatment AA, ρ = 0.54, *p* = 0.16). Similarly, there were no correlation between pen consumption during the nursery and fattening periods (treatment AA, r = −0.17, *p* = 0.52). There was no significant difference in consumption from D28 to D104 between pigs in treatment AA, with the cylinder before D28 (5.6 g/pig, CI_95%_ 3.7–7.5) and those in treatment CA, with no material provided before weaning (4.6 g/pig, CI_95%_ 3.5–5.7, *p* = 0.46). During the fattening period, from D61 to D104, consumption was higher in batch B2 (5.5 g/pig, CI_95%_ 4.5–6.5) than in batch B1 (2.2 g/pig, CI_95%_ 1.4–3.0, *p* = 0.001).

There was no need to refurnish the wood block in any pen from D28 to D104. The wood block became soiled within one or two weeks even though it was not placed on the floor.

### 3.2. Behavioral Observations

The number of pigs manipulating the enrichments (algae, wood or chain) are presented in Table 2. During the nursery period, from D28 to D60, in B1, the metal chains tended to be manipulated more often than the algae cylinder (*p* = 0.08), but this difference was not observed in B2. Up to five pigs were observed manipulating the material at the same time. No effect of the type of enrichment material was observed with regard to the number of pigs manipulating it at the same time. The number of manipulations was higher for batch B1 than for batch B2, regardless of the material: at least one pig was manipulating the material on 21% of the scans in B1 vs. 17% in B2 (*p* < 0.001). The number of marked pigs observed manipulating the algae cylinder, at least once over all the scans, was 5.0 piglets per pen (CI_95%_ 3.8–6.1), whereas it was 7.5 (6.2–8.7, *p* = 0.02) for the wood block and 7.2 (5.9–8.4, *p* = 0.03) for the metal chain. The number of individuals using the material is also different: more individual pigs used the enrichment material (A, W or MC) in B1 (8.6 pigs per pen (7.6–9.6)) than in B2 (4.5 (3.5–5.5), *p* < 0.001).

In contrast to the nursery period, the enrichment objects were used more in B2 than in B1 during the fattening period: one or more pigs were manipulating the enrichment material on 16% of the scans in B2 whereas this proportion was 4% only for B1 (*p* < 0.001). 

### 3.3. Salivary Cortisol Level

The cortisol concentration in pig saliva changed around the transfer events: it increased significantly 24 h after the transfer from the farrowing pens to the nursery pens (batch B2, Figure 2a), but this increase was not observed following the transfer from the nursery pens to the finishing pens (batch B1, Figure 2b). Nevertheless, the concentration significantly decreased between the day after the transfer (24 h post-transfer) and the following day (48 h post-transfer) in both cases. Neither sex nor enrichment treatment had significant effects.

#### 3.3.1. Health and Lesion Score

Three pigs in B1 and three pigs in B2 died from nervous or cardiac disorders. Diarrhea affected piglets in all pens in B2, with no identified relationships with the enrichment material. For lesion scoring, the frequency of score 2 (heavily injured pig) was very low for the two batches (22 observations among 2652, 0.8%). Observations with scores of 1 and 2 were thus pooled for the statistical analysis. From D28 to D61, nursery period, scratches on the front part of the body (69% of pigs) and lesions on ears (68% of pigs) were the most frequently observed lesions. After D61, in fattening pigs, the most frequent lesions were due to superficial biting, with teeth marks, or more severe biting due to chewing. Scratches were also noted on all parts of the body (from 72% to 86% of pigs affected, depending on body parts and periods). On D15, the frequency of pigs with a score of 1–2 was higher in batch B1 (156/228, 68%) than in batch B2 (53/216, 25%, χ^2^ test *p* < 0.001); this difference in lesion frequency was observed until D69 (Figure 3). A decrease in the frequency of injured pigs was observed from D30 to D37 in both batches; then the frequency of body lesions increased from D37 to D61 and remained almost unchanged up to D104. The initial lesion score at D15 had an important effect on the risk of being injured by D28: 85% of injured pigs at D15 were injured at D28 vs. 58% of non-injured pigs for B1 (odds ratio (OR) = 3.37, CI_95%_ 1.59–6.75, *p* = 0.001) and 17% vs. 10% in B2 (OR = 2.96, CI_95%_ 0.90–9.73, *p* = 0.07).This effect was thus taken into account in the analysis at D28. In batch B2, the risk for a pig having a lesion score of 1–2 at D28 was significantly lower with the algae cylinder although this object had no effect on lesion frequency in batch B1 (Table 3). No other enrichment material-related difference was found during the suckling period. During the nursery and growing periods, supplying pigs with algae cylinders, wood blocks or metal chain had no impact on the risk of body lesions.

#### 3.3.2. Weight Gain

At D15, the piglets from different treatments had not different weights (Table 4). At D28, providing piglets with algae cylinders had no impact on their weight. ADG during that period was significantly higher in Treatment A for batch B1, but this effect was not observed for B2. In B1, pigs from Treatments A and MC were significantly heavier than those from Treatment W at D61 and D104, in association with a significantly higher ADG from D28 to D61. However, an inverse effect of Treatment A was observed on B2: pigs from Treatment A weighed less than those from Treatments W and MC at D61 and D104. Pigs from Treatment A had also a lower daily weight gain from D61 to D104.

The feed conversion ratios for pigs from B1 and B2 were, respectively, 1.12 and 1.40 in the nursery and 2.39 and 2.37 for growing pigs.

## 4. Discussion

Even if done in experimental facilities, our farming conditions were very close to those of commercial farms in terms of animal management, housing, density and the number of pigs per batch and per pen. Observing animals in real farm conditions is an advantage for the implementation of the results to the daily life conditions of the pigs. Our experimental design was repeated in two successive batches, which permits to gather more robust results, but which had also showed that pig behavior can vary from one batch to another.

Before four weeks of age, the piglets ate two and a half algae cylinders at the most. This consumption rate is affordable in real farming conditions. For instance, one cylinder can be placed in the farrowing crate during the suckling period and then two cylinders in the nursery pen. Although consumption was relatively variable between the batches, the pigs nonetheless did interact with the algae cylinder during exploratory behaviors. In the farrowing crate, the algae cylinder could have been presented in a vertical position rather than a horizontal position, as it was successfully done in previous studies [8,11]. In batch B2 at D28, providing algae material reduced the risk for a pig to have a higher lesion score. Providing chewable materials in early life was previously shown as promising for reducing the severity of later tail biting [18]. However, this was not observed in our conditions. In the latter stage, the available space per pig, which was raised after transfer to the fattening pens, seems more important for preventing injuries than providing enrichment materials. Indeed, a decrease in the frequency of wounds caused by hooves when pigs walk on their congeners was observed from D30 to D37 in both batches, in association with the transfer of animals into larger nursery pens at D28. Therefore, aggressive behavior, which is correlated with wounds, did not appear to be reduced by any enrichment materials, whether they be algae cylinders, wood blocks or metal chains. The group composition in the pens remained stable during all the pigs’ life in our experiment, keeping the litter together. This situation can be different on commercial farms where pigs are mixed to obtain homogenous pen based on weight and could have reduced the risk of aggression, as has been shown [25]. Thus, the amount of aggression in the pens may have been underestimated in our conditions.

Algae material does not get lodged in between the slats of the floor of the pen and, as it is compostable, it decomposes in the manure. It is therefore compatible with slatted floor housing and its use conceivable for young pigs. However, compared with other enrichment materials, such as wood blocks and metal chains, neither of which are edible, the algae cylinders did not improve all the welfare indicators in our experimental conditions. Weaner pigs, the five- to seven-week-old pigs, in batch B1 tended to interact more frequently with the metal chain than with the gnawable objects (algae cylinders and wood blocks). Play behavior of the piglets peaks between two and six weeks of age and then progressively shifts to foraging behavior [26]. During this latter period, at five weeks of age, the metal chain seems to hold pigs’ interest more, perhaps because it moves easily and makes noise almost permanently, arousing the pigs’ curiosity. This would indicate that, contrary to previous findings, destructibility was less important for pigs than tinkling sounds [27]. Young pigs in this study interacted as much with the edible objects as they do with metal chains, which have been criticized by EFSA and considered unacceptable in other studies.

Our observations on weaners confirm collective and synchronized behavior, as described previously: up to five weaners manipulated the object simultaneously. Moreover, in half of the observed pens, more than 50% of individual pigs manipulated the object and, in a third of the pens, 100% did so. These results suggest that for a 10-pig pen, two types of enrichment objects would be better than only one, to give each pig the chance to interact with them. However, as observed by Nannoni et al. [21], objects were not homogeneously manipulated by all piglets, but mainly by some individuals.

Late-stage growers showed much higher consumption of algae material, with up to 11 cylinders for 10 pigs after 43 days. This consumption of 3 g per day and per pig is lower than the consumption of compressed straw blocks, which has been evaluated to 27 g for ad libitum fed finishing pigs [28]. However, this high consumption means that the roll dispenser must be replenished frequently, which can be time-consuming for farmers. Similar conclusions have been drawn for sawdust briquettes, which did not last more than one day because the pigs ate them readily [8]. No difference was found in grower behavior. Even if our observation periods were spread over all the fattening period, it may not have been sufficient to assess the behavior of the pigs. However, growers spent more time manipulating the equipment found in the pen, such as the trough, the door, the floor, etc., than manipulating the enrichment object. The algae cylinder, although edible, was part of the pen and did not arouse the interest of the growing pigs more than the trough or the door handle. Previous studies have shown the influence of diet on pig behavior: liquid feeding can reduce the activity level and investigatory behaviors directed towards other pigs [29] or promote unwanted behavior in terms of belly-nosing and nibbling of the ear or tail [30]. In our experiment, no difference was observed on pigs’ behavior, although the pigs of B2, which were liquid-fed, used more algae than the pigs of B1, which were dry fed. Our results, differing between batches, are in accordance with another study on the effects of different enrichment materials on pig welfare indicators [21]. This underlines the variability of the results according to batches.

Our results on salivary cortisol are in the same range as previous results [13,31,32,33]. There was no difference between treatments, although the cortisol level increased after transfer from the farrowing pen to the nursery pen. We hypothesized that algae cylinders, as a better source of stimulation than other material, would prepare the animals to cope with new stressing challenges (separation from the sow and changes in diet and environment) and to adapt faster to new situations, as suggested in other experiments [12,13,34]. The results tend to demonstrate that, in farrowing pens, the algae objects were not sufficient for piglets to enhance their ability to cope with a new situation. No difference was found after transfer from the nursery pen to the growing pens (D61 to D62). This transfer is obviously less stressing for animals than the previous one. However, due to a technical error, the pigs were fasted before transfer, which was not part of the experimental protocol. Therefore, the stress due to starvation arrived before the stress of transfer. However, no differences have been observed between the three materials (A, W or MC), even if we cannot conclude on this point, as we do not have the cortisol level before stress.

Our results show that the algae cylinder had no negative impact, neither on pig growth nor on pig health, because no disease or other health syndrome were seen specifically in the pigs furnished with these cylinders during the study. The cylinders were composed of carbohydrates (alginic acid, laminarine, mannitol and cellulose) and minerals. No consistent growth differences were found between pigs with and without this material. Our results are in accordance with previous results, which found no growth performance differences between pigs given the different kind of biodegradable objects [13,21]. However, even if the cylinders represented only a tiny part of the pigs’ consumption, they can bring diversity in the feed intake and this could have resulted in better microbiota, for instance. However, if a routine use is envisaged, the nutritional quality of the material should be carefully checked as well as the absence of toxicity, due, for example, to the presence of heavy metals, which could be transmitted to humans through the consumption of pork meat. The growth of the pigs in this experimental farm is very high and thus difficult to improve.

To better assess the potential benefits of the algae cylinders, they should be tested in suboptimal farming conditions, i.e., with poor performance, with groups of pigs mingled at different stages of their life and with pigs that would be neither tail-docked nor teeth-clipped.

## 5. Conclusions

Behavioral observations showed that, in the conditions of this study from D15 to D104—so, during almost all their life—pigs interacted with the algae cylinders, thus expressing interest in this enrichment material. However, from our study results, algae cylinder seemed as attractive for pigs as metal chains, although the two materials are classified in different categories of enrichment material by the EU Commission (of suboptimal and marginal interest, respectively). However, this material is an interesting enrichment to be used as a potential sub-optimal material, providing an opportunity for farmers to diversify their enrichment materials in the context of national regulations related to enrichment, particularly in case of a tail biting outbreak, in an attempt to suppress this redirecting behavior.

## Figures and Tables

**Figure 1 animals-11-00315-f001:**
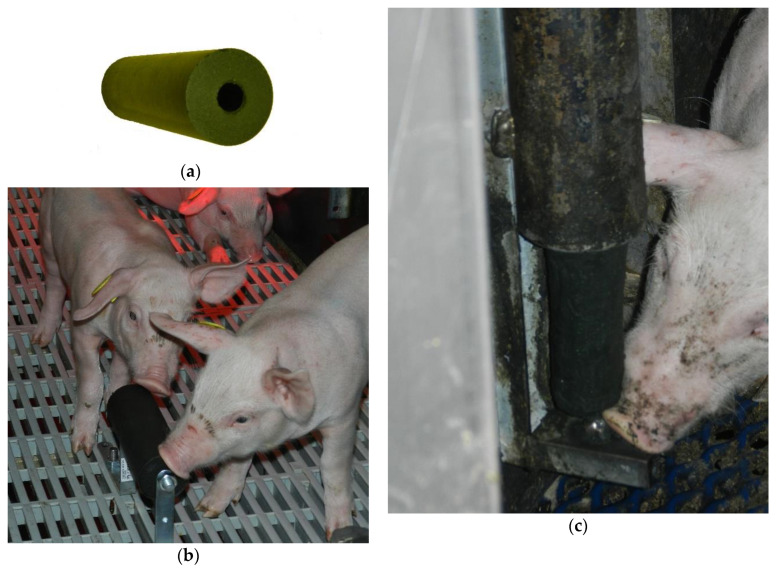
Enrichment materials: (**a**) algae material made of brown algae *Saccharina latissimi*, post-harvest dried and compressed under heat into a cylinder (15 × 5 cm, 380 g); (**b**) horizontal algae roll dispenser in the farrowing pens; (**c**) vertical algae roll dispenser in the post-weaning and fattening pens.

**Figure 2 animals-11-00315-f002:**
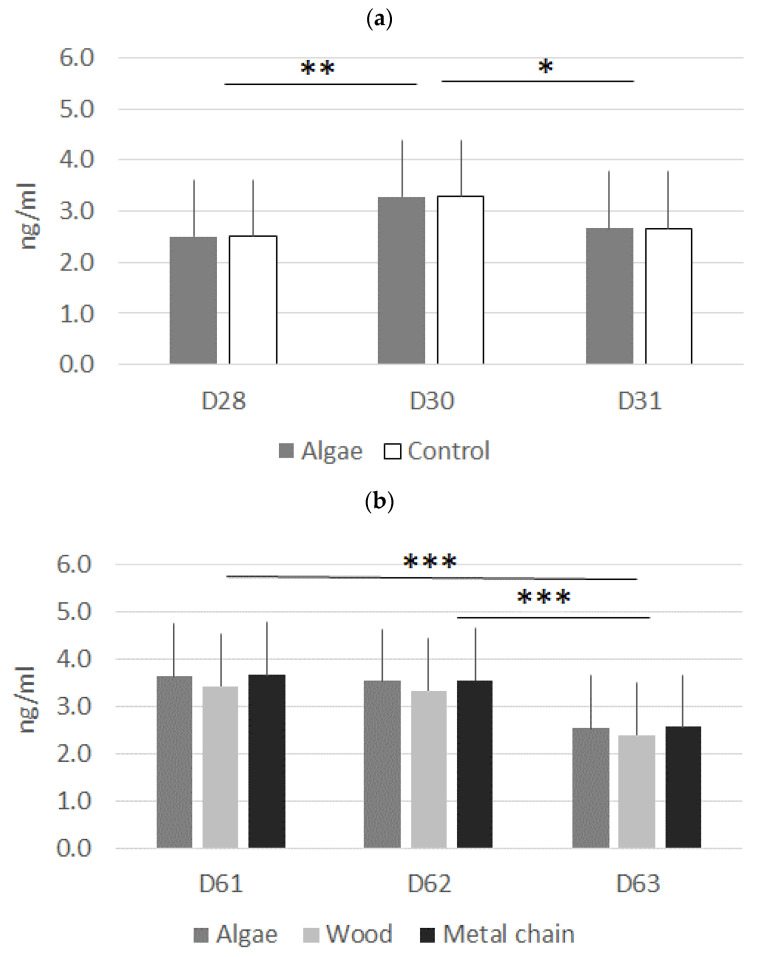
Cortisol concentration in pig saliva according to the enrichment material provided before and after transfer (**a**) from the farrowing pens to the nursery pens (batch B2, *n* = 52 piglets); and (**b**) from the post-weaning pens to the finishing pens (batch B1, *n* = 156 pigs). Results are shown as least square means ± standard error. A mixed ANOVA model with the pen as random effect was used to compare cortisol levels (* means *p* < 0.1; ** means *p* < 0.01; and *** means *p* < 0.001).

**Figure 3 animals-11-00315-f003:**
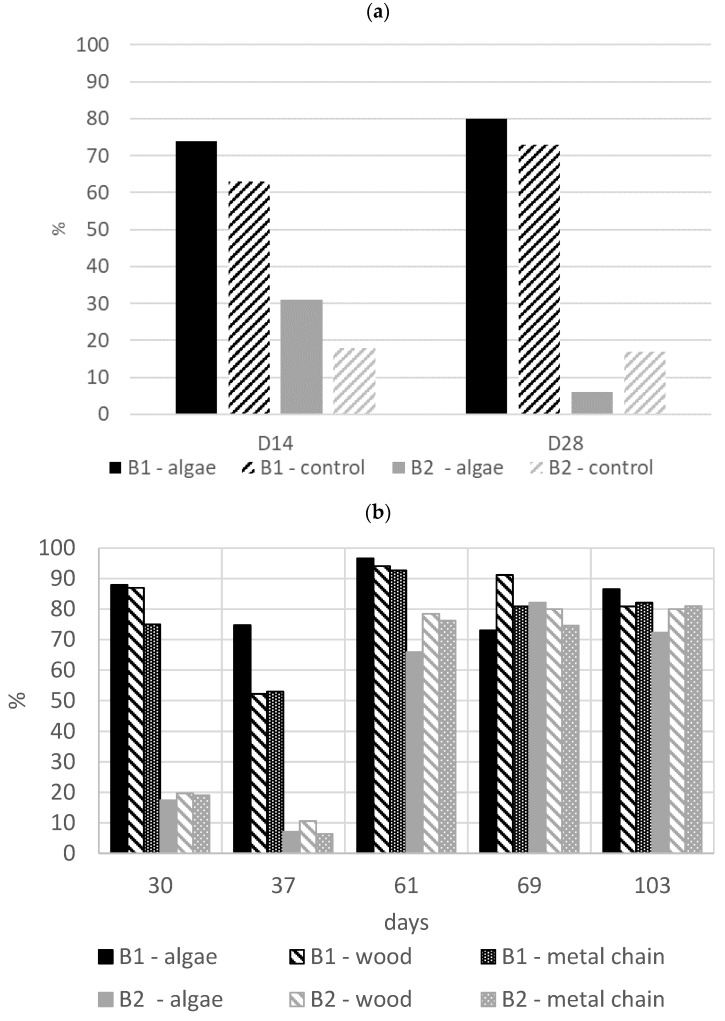
Frequency of pigs with a lesion score of 1–2 (1: moderately and 2: heavily injured) according to enrichment material from (**a**) Day 15 to Day 28; and (**b**) Day 30 to Day 103 (*n* = 128 in batch B1 and *n* = 114 in batch B2).

**Table 1 animals-11-00315-t001:** Experimental design and types of enrichment. Pigs from one litter remained together in the same group and in the same pen during the whole study period. D: day of experiment; B: batch of pigs.

Farrowing Pen (D15 to D28)	Nursery (D28 to D61) and Growing Pens (D61 to D104)	Enrichment Material
Algae B1: 10 litters (113 piglets) B2: 10 litters (108 piglets)	Algae B1: 4 pens (43 piglets, 16 females and 27 males) B2: 4 pens (44 piglets, 22 females and 22 males)	AA ^1^
	Wood B1: 3 pens (36 piglets, 20 females and 16 males) B2: 3 pens (34 piglets, 19 females and 15 males)	AW ^1^
	Metal chain B1: 3 pens (34 piglets, 13 females and 21 males) B2: 3 pens (30 piglets, 16 females and 12 males)	AC ^1^
No enrichment B1: 10 litters (115 piglets) B2: 10 litters (108 piglets)	Algae B1: 4 pens (48 piglets, 25 females and 23 males) B2: 4 pens (42 piglets, 20 females and 22 males)	CA ^1^
	Wood B1: 3 pens (33 piglets, 11 females and 22 males) B2: 3 pens (32 piglets, 16 females and 16 males)	CW ^1^
	Metal chain B1: 3 pens (34 piglets, 22 females and 12 males) B2: 3 pens (34 piglets, 17 females and 17 males)	CC ^1^

^1^ Type of enrichment in farrowing pen (first letter) and after weaning in nursery and fattening pens (second letter): AA, algae-algae; AW, algae-Wood; AC, algae-Metal Chain; CA, no enrichment-algae; CW, no enrichment-wood; CC, no enrichment-Metal Chain.

**Table 2 animals-11-00315-t002:** Number of pigs (least square mean ± standard error, total number of observed pigs in brackets) manipulating the enrichment material per scan according to the material during the nursery period (Day 28 (D28) to D60 (*n* = 11.823 scans)) and during the fattening period from D70 to D84 (*n* = 1800 scans). The comparison of the material treatments was made with a mixed ANOVA model with the pen as a random effect (*p* value). As a batch effect was detected, batches were analyzed separately.

Pig Age	Enrichment Material	Batch B1		Batch B2	
D28–D60	Algae cylinder (4 pens)	(*n* = 91) 0.21 ± 0.03	*p* = 0.08	(*n* = 86) 0.22 ± 0.04	*p* = 0.95
	Wood block (3 pens)	(*n* = 69) 0.25 ± 0.05		(*n* = 64) 0.23 ± 0.05	
	Metal chain (3 pens)	(*n* = 68) 0.33 ± 0.04		(*n* = 64) 0.23 ± 0.03	
D70–D84	Algae cylinder (4 pens)	(*n* = 91) 0.05 ± 0.02	*p* = 0.98	(*n* = 86) 0.24 ± 0.08	*p* = 0.77
	Wood block (3 pens)	(*n* = 69) 0.04 ± 0.02		(*n* = 64) 0.18 ± 0.07	
	Metal chain (3 pens)	(*n* = 68) 0.05 ± 0.01		(*n* = 64) 0.25 ± 0.05	

**Table 3 animals-11-00315-t003:** Risk of being injured (odds ratio (confidence interval at 95%)) by Day 28 (D28) at weaning (*n* = 228 pigs in batch 1 and *n* = 215 pigs in batch 2) and during the nursery and finishing periods, D28 to D104 (*n* = 228 pigs in batch 1 and *n* = 215 pigs in batch 2), according to the enrichment material. The 95% confidence intervals were calculated from the logistic model for the mixed data, taking into account repeated measures during the post-weaning and finishing periods.

Pig Age	Enrichment Material	Batch B1		Batch B2	
D28	Algae Control ^1^	1.14 (0.41–3.21) 1	*p* = 0.80	0.18 (0.05–0.69) 1	*p* = 0.01
Day28 to Day104	Algae Wood Metal chain^1^	1.36 (0.41–4.51) 1.30 (0.36–4.70) 1	*p* = 0.62 *p* = 0.68	0.90 (0.39–2.11) 1.38 (0.55–3.48) 1	*p* = 0.81 *p* = 0.49

^1^ Reference treatment.

**Table 4 animals-11-00315-t004:** Weight (kg) and average daily gain (ADG, g/day) (least square mean ± standard error) according to type of enrichment material in batch 1 (B1; *n* = 228 pigs) and batch 2 (B2; *n* = 215 pigs) during the suckling period, from Day 15 (D15) to D28, during the post-weaning period, from D28 to D61, and during the fattening period, from D61 to D104. Comparison was made with a mixed ANOVA model with the pen as random effect. As a batch effect was detected, batches were analyzed separately.

Pig Growth Performance	Enrichment Material	Batch B1		Batch B2	
Weight D15	Algae Control	4.59 ± 0.13 4.69 ± 0.20	*p* = 0.67	5.12 ± 0.14 5.19 ± 0.19	*p* = 0.79
Weight D28	Algae Control	9.42 ± 0.23 9.05 ± 0.32	*p* = 0.34	9.39 ± 0.19 9.54 ± 0.29	*p* = 0.67
ADG D15–D28	Algae Control	339 ^a^ ± 7 313 ^b^ ± 9	*p* = 0.02	325 ± 7 332 ± 9	*p* = 0.57
Weight D61	Algae Wood Metal chain	28.8 ^a^ ± 0.7 26.4 ^b^ ± 0.6 28.4 ^a,b^ ± 0.7	*p* = 0.02	25.4 ^a^ ± 0.7 28.6 ^b^ ± 0.6 28.3 ^b^ ± 0.5	*p* = 0.01
Weight D104	Algae Wood Metal chain	70.6 ^a,b^ ±1.2 67.7 ^a^ ±1.1 71.1 ^b^ ± 0.7	*p* = 004	59.8 ^a^ ± 0.8 63.4 ^b^ ±1.0 62.8 ^b^ ± 0.7	*p* = 004
ADG D28–D61	Algae Wood Metal chain	557 ^a,b^ ± 16 539 ^b^ ± 12 585 ^a^ ± 15	*p* = 0.03	503 ^a^ ± 15 570 ^b^ ± 12 553 ^b^ ± 12	*p* = 0.02
ADG D61–D104	Algae Wood Metal chain	993 ± 15 982 ± 19 1021 ± 11	*p* = 0.13	815 ± 14 825 ± 11 827 ± 13	*p* = 0.79

‘Letters “a”, “b” and “c” mean that least square mean are significantly different with *p* < 0.05.

## Data Availability

The data presented in this study are available on request from the corresponding author.

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
