# Peer review of "Compressed Brown Algae as a Potential Environmental Enrichment Material in Growing Pigs"

_animals, 2021, doi:10.3390/ani11020315_

Round 1

Reviewer 1 Report

 BRIEF SUMMARY

The aim of this study is to test the effect that an enrichment material made from brown algae (Saccharina latissimi) can have on pig health, performance and body condition. It is an interesting paper, because it searches for an enrichment material that can be categorised as optimal, that is, edible, chewable, investigable and manipulable. The obtained results indicate that this material does not significantly improve pig welfare when compared with other commonly used elements, such as metal chain.

The study question is original and clearly defined. However, not every conclusion is based on the results, and the results themselves do not provide a major progress in current knowledge.

BROAD COMMENTS 

Council Directive 2008/120/EC states in Annex I that neither tail-docking nor reduction of corner teeth must be carried out routinely but only where there is evidence that injuries to sows’ teats or to other pigs’ ears or tails have occurred. Furthermore, it stablishes that, before carrying out these procedures, other measures shall be taken to prevent tail-biting and other vices, taking into account environment and stocking densities.

Moreover, the Commission Recommendation (EU) 2016/336 stablishes minimum standards in order to prevent pigs from directing their exploratory behaviours and manipulation towards others and thus prevent tail-biting and promote pig welfare. It recommends, among other possibilities, providing adequate enrichment materials.

Contrary to what is laid down in the above-mentioned Directive, this paper states that all pigs were tail-docked and teeth-clipped within the first three days of life (lines 96-97). Accordingly, the potential injuries they may cause on each other are really conditioned and limited due to this fact. Besides, the experiment was done in experimental facilities, where farming conditions are determined by the researchers, so these practices should not have been carried out. Obviously, the fact that all pigs were tail-docked and teeth-clipped considerably conditions the results of the study and that is probably why the results do not significantly improve pig welfare.

SPECIFIC COMMENTS 

line 71. The authors claim that the algae cylinder can be considered as a sub-optimal material, although, according to the categorisation included in the Commission Recommendation (EU) 2016/336, this material could be considered as optimal. Why do the authors consider it sub-optimal? Which characteristic or characteristics does it fail to comply with?

Lines 75-76. This statement should include references.

Lines 84-98. The section “animals and housing conditions” lacks accuracy when describing housing, as pen sizes are not specified and, therefore, neither are animal density per housing or available air volume. Both aspects would be relevant when researching tail-biting. Environmental conditions, such as humidity or temperature, are also not specified.

Table 1. Why does the number of piglets in farrowing pens not match those in nursery and growing pens? 228/221 and 216/223. If the difference is caused by losses when treating with algae, this should be specified. Besides, the number of animals increases in the second phase of the study for the treatment with no enrichment.

Lines 166-167. Why is average daily gain not calculated for the fattening period, from D61 to D104?

Lines 224-226. Cortisol concentration in pig saliva is increased in weaning, because this is a sensitive phase characterised by simultaneous stresses including separation from the sow, mixing of litters, and changes in diet and environment. This considerably increases stress on the animals. Obviously, this situation does not occur when the animal is transferred from nursery pens to finishing pens.

Line 251. 53/214 is mentioned, when it actually should be 53/216.

Lines 319-320. The authors claim that “in latter stage, available space per pig seems more important for preventing injuries than providing enrichment materials”. However, the available space per pig is never quantified, which would enhance the discussion.

Lines 364-366. This statement casts reasonable doubts, as already explained in the comment regarding lines 224-226.

Lines 382-390. Conclusions are very general and should be rewritten. Some of them are not based on the results obtained in this study, for example “this material […] could also be used as a support to bring nutriment”.

Reviewer 2 Report

1. General comments

This is a very up-to-date topic. The prevention of tial biting in growing pigs has been the focus of much research on environmental enrichment. Therefore, the goal of the manuscript is supportable. However, the manuscript needs to be supplemented and clarified in many places.

2. Detailed comments

  1. I propose to divide the chapters of Materials and Methods, Results, Discussion into two parts: 1. Pre-trial made it exparimental conditions (see: line 75-76). This study, based on biological physiological examinations, is very important to demonstrate the the test material is safe and suitable for use in pig operations. These examinations cannot be left out in the case of scientific paper. 2. Field trial: this part should be content the submitted manuscript.
  2. Line 84: The genetic background of the pigs must be described (this is also information on the stress tolance of the animals used in the study).
  3. Line 87-88: More informations are needed on diets used in the study (composition of the diets, calculated and/or analysed value of the energy and nutrient contetnts. Feed additives if any.
  4. Line 95: What was the sex ratio in the pens?
  5. Line 100-105: The chemical composition of the algae material must be given. Since the test material is a by-product of the cosmetics and food industry (see lines 71-72), the chemical composition of the test substance (enrichment material) and its stability is very important information for the production of the safe and high quality animal origin foodstuffs (e.g. meat). Is there any information on possible heavy metal contamination?
  6. Line 114: Better quality (higher resolution) images are required. Not too much can be seen on Figure 1a,b and c.
  7. Line 137-139: Why aren't the morning and afternoon video recording time the same?
  8. Line 166-167: The daily feed intake (DFI: kg/d) and the feed conversion ratio (FCR: kg feed/kg gain) are important to show for the Reader. (Due to group housing, of course, there will be only one data/pen.) It is well known that these parameters are also change under stress. But they may also be affected by increases in physical activity.
  9. Line 263-269: Figure 3: Because of long time ellapses between each data recording, the representation of the data with a line graph is incorrect. This is because due to the long time interval between two observations, there may not be a linear effect. Therefore, it is not correct to connect the points with a line. I suggest to presenting the data on bar chart. In addition to the mean values please indicate the s.d. or s.e. values as well.
  10. Line 285: Data on daily feed intake and feed conversion ratio are needed (see remarks in number 9).
  11. Line 295-297: This sentence belongs to the aim of the study. This should be not a part of discussion. Please omit this sentence.
  12. Line 297-302:This sentence belongs to the section of Materials and method. Please omit this sentence.
  13. Line:303-305: Please omit this sentence. In this experiment this was not the subject of the study. Therefore, this does not to be discussed in this chapter.
  14. Line 306-307: Please omit here this sentence. This sentence has the right place not in this chapter. This must be inserted in the chapter Introduction.
  15. Line 367: Explanation is needed, why the pigs were fasted before transfer if is was not part of the protocol.
  16. Line 382-386: Based on own experimental data, a clear conclusion should be drawn: whether the authors recomment the use of algae sylinders in pig operations (pig farms).
  17. Line 388-390: "Algae cylinders could also be used as a support to bring nutrients". This is not cleare to me why was drawn this conclusion: in this field trial there were not carried out such measurements, fordermore, the chemical composition of the algae cylinder and the algae intake of the pigs are not known.
  18. Line 2-3: I suggest to change the title of the manuscript: the current title do not express what the manuscript is about. The title should indicate that in a comperative study a testing with growing pigs was carried out with an environmental enrichment material based on brown algae.

Reviewer 3 Report

General comment

This is a nice study but it needs to be further improved before publication.

In particular, you mention that the main aim of the study is to assess the attractiveness of the algae cylinder but your behavioural observations are very limited (especially in the fattening period) and you did not do a preference test, which would have been a gold standard measure for attractiveness!

Then, I am not sure you can talk about “replicates” when you changed one of the feeding conditions (they would appear to be different, moreover in a study where you looked at weight gain and consumption of an edible object).

You will find hereunder specific comments on the manuscript.

Introduction

General: I would have expected an explanation on what makes the algae suboptimal and why you choose a sub-optimal object rather than an optimal one...

L52: “such as roughage”

L76: “on an experimental farm”

L80: “suitable for slatted floors”

Material and Methods

L121: “during the whole trial”

L126: suggest modifying the sentence slightly: “…into the three enrichment treatments: algae, wood and metal chain (CA, CW and CMC; respectively)”

L136-143: So if I understand correctly you did individual observations during the nursery period but not during the fattening period? What about the suckling period?

I imagine the cameras could not be installed in the fattening pens, but that would have been better to identify the pigs in the fattening period too in order to link individual data through the trial (or if you collected individual data, then make sure to mention it and how the individual identification was done).

Also, why have you reduced the time of observation so dramatically in the fattening period ? It seems to me that observing 1.5h on 4 days is limited to acquire data… can you justify your choice?

Finally, what behaviours did you observe? You mention “foraging” and “exploratory” as well as “play” later, so did you make the difference between all of these behaviours when observing them?

L144-155: Wy did you choose to split the collection of saliva? It would have been better to collect it on the same days for both batches in order to make comparisons.

L170: can you detail which test was used for which data?

Results

L204: may be make clear you refer to any of the material provided (chain, wood, algae).

L210-211: this contradicts what you stated at L204…

L213: you refer to “fattening” before, check your manuscript for consistency

Figure 2: is the figure to the resolution quality expected by the journal (1200 dpi) ? It looks blurry.

L248-249: how can you infer the scratches to be caused by pigs walking on each other? Is it from your observations (i.e. did you witness scratches appearing after a pig walked on another?). That statement needs back-up and should be in the discussion rather than in the results section.

Figure 3b: Hard to distinguish wood and Algae, maybe use dotted lines to make the difference clearer? Also check the quality (resolution) of the figure.

Table 3: Explain in the figure caption why Control and Metal Chain do not have confidence intervals (I guess it is because they were the reference in the analysis?).

Table 4: Weight D104, Batch B2 : please check the letters, as I am not sure it is possible to have a significant difference between 59.8±0.8 and 62.8±0.7, but not between 59.8±0.8 and 63.4±1.0

L292: “with different superscript letters….”

Discussion

L298: rearing conditions or husbandry – breeding usually refers only to producing offspring (i.e. genetics, mating, gestation).

L301: remove “twice”, “repeated in two successive batches is enough)

               “which permits

L307-308: What makes the algae suboptimal? Also, one could question that “suboptimal” is an opportunity: everyone should aim for optimum, not suboptimum…

L308: I would rather refer to the age of piglets or the time to eat the cylinders.

L309: I am not sure “clearly” is the appropriate term… Maybe “for instance/example”?

L312-313: I am not sure I understand your point, do you mean that the cylinder would attract the same attention if placed vertically or horizontally? If so, please make is clearer in your statement.

L315-317: yes but you compare your results to a very different set-up: algae are edible and the objects of Yang et al. were not! I do not see the point to compare the two and I would delete it (you make a valuable comparison after with the study of Telkanranta et al.)…

L319-320: reference for that statement?

L323: after this paragraph I would have expected a discussion on the effect of keeping the litter together on lesions. First of all, this is not a standard procedure on commercial farms (they usually mix pigs to obtain homogenous pen based on weight), and secondly that had been shown to reduce the risk of aggression:

  • Rydhmer, L., Hansson, M., Lundström, K., Brunius, C., & Andersson, K. (2013). Welfare of entire male pigs is improved by socialising piglets and keeping intact groups until slaughter. Animal, 7(9), 1532-1541.

L335: there are some extra spaces in this line (after “EFSA”, and “ , “)

L344: yes but for how many pens and how much time? This is what matters!

L349-350: maybe criticize the fact that you observed them only 1.5h on four days (6h across 45 days)!

L368: replace “Then” by “Therefore” as you are inferring a consequence to the starvation and delete “there” (“the differences that could have been observed…”)

I am surprised by this statement… if you knew the pigs were starved before transfer, then why did you not choose different days to observe? I did not find it clear in your Material and Methods that pigs were observed just after moving, but that is also a flaw in your study design: if you wanted to assess the attractiveness of the material then why did you choose to observe behaviours on the most disturbed/stressful days instead of days were pigs are settled??? That really does not make sense to me…

L373-374: Where are these 11 cylinders coming from? Is it the extreme? Because you refer to it twice in the discussion as if it was the “normal” consumption, but in your results you mention an average of 3.5±1.5 cylinders per pen for the whole fattening period… This needs to be clear to not mislead the reader into false conclusions!

L377: Yes, I guess this is the case for most modern breeds… however the cylinder brought diversity in their feed intake and this could have resulted in better microbiota for instance.

L380: delete one of the period (“..”)

Conclusions

L387: did you differentiate between foraging and other exploratory behaviours? To me it sounds quite logical that a stationary elevated object did not improve foraging, since this behaviour in pigs requires extensive search for food by rooting the ground… your object did not allow any features of natural foraging in pigs!

Round 2

Reviewer 1 Report

Dear Authors,

All my suggestions have been considered and I agree with all changes made.

Best wishes,

Author Response

Dear Authors,

All my suggestions have been considered and I agree with all changes made.

Best wishes,

Response: we thank the reviewer for her/his help in improving the manuscript.

Reviewer 2 Report

The manuscript has been improved a lot after revision. I accept the most of the authors' response. I suggest adding to the text two brief additions only:

Line 87-88: The energy (DEs, or MEs and nutrient (e.g. C. protein, C.fat, C. fiber, Dig. LYS, Dig M+C, Ca, P) content of the diets would be very important information for the reader, whereas this may indicate that the animals' energy and basic nutrient supply was appropriate for their age.

The other possibility is as follows: Energy and nutrient supply of piglets met NRC (2012) or INRA (????) recommendation.

Line 166-167 and Line 285: I suggest indicating in the text the feed intake was recorded on half room level (included five pens as one unit)

Author Response

The manuscript has been improved a lot after revision. I accept the most of the authors' response. I suggest adding to the text two brief additions only:

Line 87-88: The energy (DEs, or MEs and nutrient (e.g. C. protein, C.fat, C. fiber, Dig. LYS, Dig M+C, Ca, P) content of the diets would be very important information for the reader, whereas this may indicate that the animals' energy and basic nutrient supply was appropriate for their age.

The other possibility is as follows: Energy and nutrient supply of piglets met NRC (2012) or INRA (????) recommendation.

Response: As the food composition is in line with usual food practices, we choose the second suggestion of the reviewer and cite both following references of INRA (lines 100-102 and Lines 491-494).

Line 166-167 and Line 285: I suggest indicating in the text the feed intake was recorded on half room level (included five pens as one unit)

Response: we mentioned this data recording in the Material&Method and Results sections (lines 175-177 and 300-301).